# Molecular design principles of Lysine-DOPA wet adhesion

Yiran Li[1,2,4], Jing Cheng[1,4], Peyman Delparastan [1,4], Haoqi Wang [2,4], Severin J. Sigg[1], Kelsey G. DeFrates [1], Yi Cao [2] & Phillip B. Messersmith [1,3]✉

The mussel byssus has long been a source of inspiration for the adhesion community. Recently, adhesive synergy between flanking lysine (Lys, K) and 3,4-Dihydroxyphenylalanine (DOPA, Y) residues in the mussel foot proteins (Mfps) has been highlighted. However, the complex topological relationship of DOPA and Lys as well as the interfacial adhesive roles of other amino acids have been understudied. Herein, we study adhesion of Lys and DOPA-containing peptides to organic and inorganic substrates using single-molecule force spectroscopy (SMFS). We show that a modest increase in peptide length, from KY to (KY)$_3$, increases adhesion strength to TiO$_2$. Surprisingly, further increase in peptide length offers no additional benefit. Additionally, comparison of adhesion of dipeptides containing Lys and either DOPA (KY) or phenylalanine (KF) shows that DOPA is stronger and more versatile. We furthermore demonstrate that incorporating a nonadhesive spacer between (KY) repeats can mimic the hidden length in the Mfp and act as an effective strategy to dissipate energy.

[1] Departments of Bioengineering and Materials Science and Engineering, University of California, Berkeley, CA, USA. [2] Department of Physics, Nanjing University, 210093 Nanjing, P. R. China. [3] Materials Sciences Division, Lawrence Berkeley National Laboratory, Berkeley, CA, USA. [4]These authors contributed equally: Yiran Li, Jing Cheng, Peyman Delparastan, Haoqi Wang. ✉email: philm@berkeley.edu

One of the great challenges faced by man-made adhesives is binding in the presence of water, salts, and surface contaminants[1]. Marine mussels, on the other hand, have perfected the art of adhering tenaciously to a variety of surfaces in wet conditions[2]. The strong attachment of mussels is mediated by the byssus, a proteinaceous holdfast that is formed by secretion and solidification of specialized adhesive proteins[3–6]. A unique feature of these interfacial proteins is the presence of large amounts of post-translationally modified amino acid 3,4-dihydroxyphenylalanine (DOPA), a catechol-containing residue that is believed to be a major contributor to wet adhesion[7–11]. Bioinspired design principles based on mimicking these interfacial proteins have been employed extensively and resulted in a variety of catechol functionalized polymers for bio-compatible adhesives, self-healing hydrogels, and surgical wound closure materials[12–22].

Nevertheless, the true potential of mussel-inspiration may not be fully realized until the hidden complexities in the structure and biofabrication of these adhesive proteins is revealed. For instance, a large number of positively charged Lys or arginine (Arg) residues are found in proximity to DOPA along the protein backbone[23]. Recently, Maier et al. and others utilized surface forces apparatus (SFA) to study the adaptive synergy between amine and catechol in binding to wet mica, using small molecule cyclic analogs with Lys or Arg present adjacent to catechol or phenyl groups[1,24,25]. Their results showed that adhesion energy is remarkably higher when both catechol and amine are present, suggesting a synergistic effect between these functional motifs. SMFS studies further revealed that the average detaching force for DOPA and Lys dipeptide is ~300 pN on mica surface, which was observably higher than that of a dipeptide in which the Lys side chain was protected (~90 pN)[9,26]. Although these studies indicated an amino-catechol synergy, the influence of other amino acids, peptide length and topology on adhesion and cohesion still remains unclear.

Interestingly, in recent studies a DOPA-deficient foot protein from green mussels was shown to possess strong wet adhesion capabilities[27]. Results of SFA measurements on Phe and Lys model peptides by Gebbie et al. indicated that adhesion of these peptides to mica exceeded even that of DOPA-containing analogs[5]. The surprisingly strong adhesion of these peptides was attributed to the interaction of Lys with mica surface as well as intermolecular cation-π cohesion between Lys and Phe residues. These provocative results challenged the notion that catechols are required for wet adhesion and sparked the design of Phe-based synthetic adhesives[5,28,29].

Although SFA measurements elegantly revealed the important roles of DOPA, Lys and Phe in wet adhesion and cohesion, the ensemble nature of SFA experiments precludes accurate determination of the molecular mechanism and provide indirect evidence for the adhesive functions of Lys and Phe. Since incorporation of DOPA into polymers and peptides has proven to be more synthetically demanding than Phe, the enticing possibility of employing Phe instead of DOPA in bioinspired molecular designs motivates further studies of the interfacial adhesion properties of Phe, alone or in combination with Lys. Furthermore, in Mfp-5, Lys and DOPA often appear in Lys-DOPA-Lys symmetric structure with several additional amino acids located between these sites instead of the adhesive DOPA (Fig. 1a). However, the effect of binding site density and topological structure on adhesion of Mfps and synergistic effects between DOPA and Lys residues is yet to be fully understood.

Here, we probe the single molecule adhesive behavior of Mfp-5 analog peptides of various length and composition on organic and inorganic substrates (Fig. 1b). We first measure the detaching force of DOPA-Lys peptides of various lengths (KY), (KY)$_3$, and (KY)$_{10}$ with TiO$_2$. Next, we use SMFS to quantitively compare the strength of interaction of the (KF)- and (KY)-containing peptides with different surfaces and evaluate importance of cation-π mediated binding in (KF). We then study interfacial adhesion strength of a synthesized analog peptide of Mfp-5. Furthermore, by inserting short polyethylene glycol oligomers (P8) between KY repeat units, we mimic the hidden length in Mfps and reveal the important function of this hidden length in wet adhesion. Our results shed light on the interplay between chemical sequence and topological structure in the mussel adhesive proteins and provide a solid framework for rational design of bioinspired wet adhesives.

## Results

**Peptide synthesis and SMFS measurements**. All peptides were synthesized via Fmoc-strategy on solid phase (Fig. 2) with an azide-(PEG)$_6$-COOH linker conjugated to the C-terminus and then covalently attached to 5 kDa PEG-modified AFM cantilevers via Cu-free click chemistry for use in the force spectroscopy measurements (Supplementary Methods, Supplementary Figs. 1–8). The AFM-based SMFS approach has been widely used to measure adhesion of biomolecules and polymers and to analyze elastic protein topological structure, ligand recognition and polymer mechanics[30–35]. In a typical force spectroscopy experiment, the cantilevers were approached to the substrate at a constant speed of 1000 nm s$^{-1}$, held on the surface with a constant force of 0.3–0.5 nN for 2 s, and then retracted at the same speed. In ~2–5% of total force-extension (F–X) curves, single rupture force events were observed at ~20–60 nm, indicating the peptide surface detachment (Supplementary Fig. 9). Each individual peak was fitted with the worm-like chain (WLC) model (red line), using a persistence length range of 0.36–0.40 nm, consistent with the PEG linker and suggesting the rupture of single-peptide detachment. Rupture events with persistence lengths either larger than 0.40 nm or smaller than 0.32 nm were discarded in the data analysis (Supplementary Fig. 10).

We first studied effects of number of the KY repeating units on the adhesion of mussel-inspired peptides. For this purpose, we synthesized peptides (KY), (KY)$_3$, and (KY)$_{10}$ and measured their adhesion against TiO$_2$. Representative F–X curves and histograms of rupture force distribution for the interaction between the peptides and the surface are shown in Fig. 3. For the (KY) peptide, the adhesion force distribution showed a narrow dominant peak located at ~120 pN and a less probable wide distribution from 200 to 900 pN (Fig. 3a). The dominant peak located at lower force matches previously reported values and could mainly result from the rupture of individual DOPA–surface interactions[9,36,37]. The second broad peak may be attributed to the synergistic binding of (KY) to the TiO$_2$. Due to the surface roughness, pulling direction and molecular dynamics, this synergistic effect cannot be successfully established in every pulling cycle to the same extent, leading to a broad distribution in the rupture force values[26,38–40]. For the longer peptide (KY)$_3$, the rupture force distribution showed only a major peak at ~300 pN. Compared to the (KY) peptide, the (KY)$_3$ sequence has more positively charged Lys residues which increases the strength of the coulombic charge interactions with the negatively charged TiO$_2$ surface. The increased positive charges can also result in a more effective removal of the hydration layer on the surface and facilitate the binding of DOPA to the substrate[1]. Moreover, in (KY)$_3$, the repetitive sequence allows the formation of a symmetric KYK structure, which could further promote the synergistic binding and make it more adaptable and versatile. As a consequence, the adhesion strength is enhanced and the detaching force increased to ~300 pN (Fig. 3b). Although most of

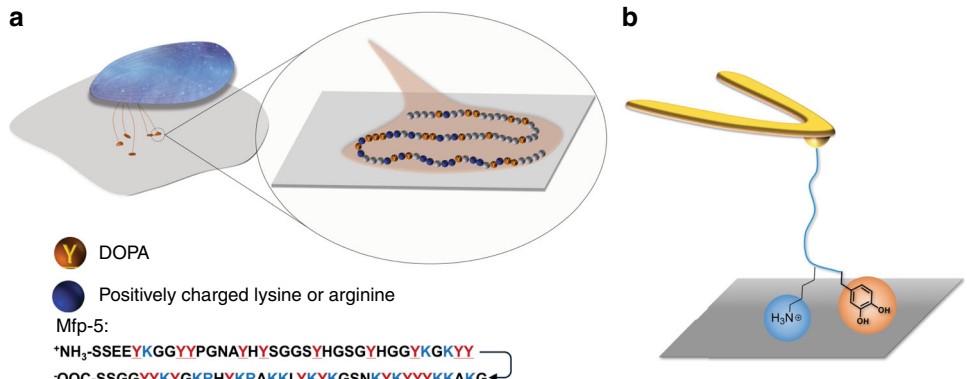

**Fig. 1 DOPA and Lys in marine mussel adhesion. a** Schematic of the mussel thread attachment to the surface with zoom-in showing the Mfp-5 sequence and **b** schematic of the SMFS experiments to measure the strength of interaction of the peptides with the organic and inorganic substrates.

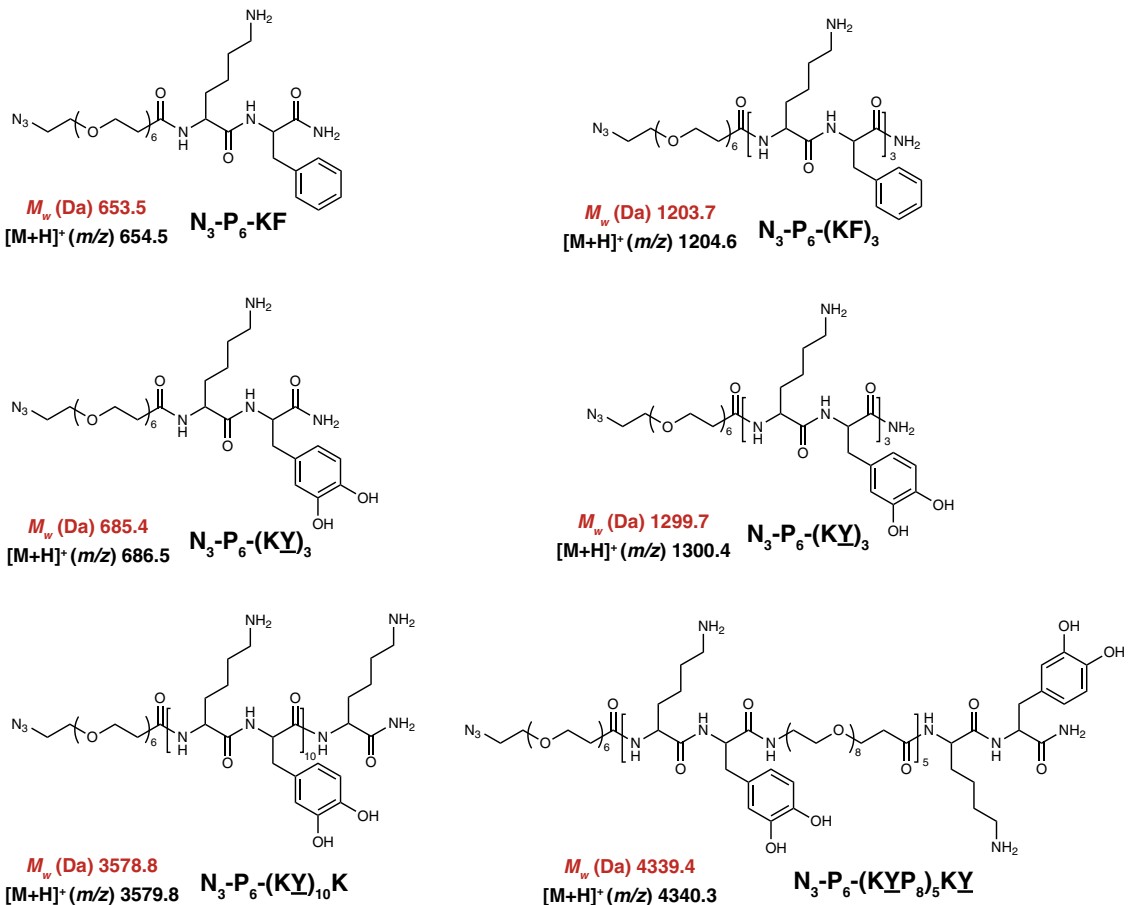

**Fig. 2 Chemical structure of the synthesized peptides.** Calculated exact mass and measured mass (obtained from MALDI) are shown in red and black, respectively.

the F–X curves only exhibited one rupture peak, in some rare cases (<5%) we were able to observe two or three distinct rupture peaks in one F–X curve which might result from sequential detachment of individual (KY) units from the surface (Supplementary Fig. 11). The contour length increment of these peaks is ~1 nm, consistent with the distance between repeat units in the peptide. The low observation rate of multiple detachments is likely a result of approaching the AFM detection limit, and more importantly conformational limitations that prevent all the adhesive moieties from binding effectively to the surface.

To study the cooperative binding of KY units in greater depth we synthesized (KY)$_{10}$ peptides for SMFS measurements.

Although it is generally expected that more adhesive units located along the peptide chain should lead to higher adhesion strength, to our surprise, simply incorporating more adhesive motifs into the structure did not enhance adhesion performance remarkably. Interestingly, the majority of the F–X curves only contained one or two rupture peaks with an average detaching rupture force of ~250 pN (Fig. 3c and Supplementary Fig. 12) similar to that of the (KY)$_3$ peptide. For the case of (KY)$_{10}$ peptide, backbone rigidity, peptide conformation, as well as surface roughness and hydration might possibly make it highly unlikely for all the (KY) units to simultaneously interact with the substrate effectively.

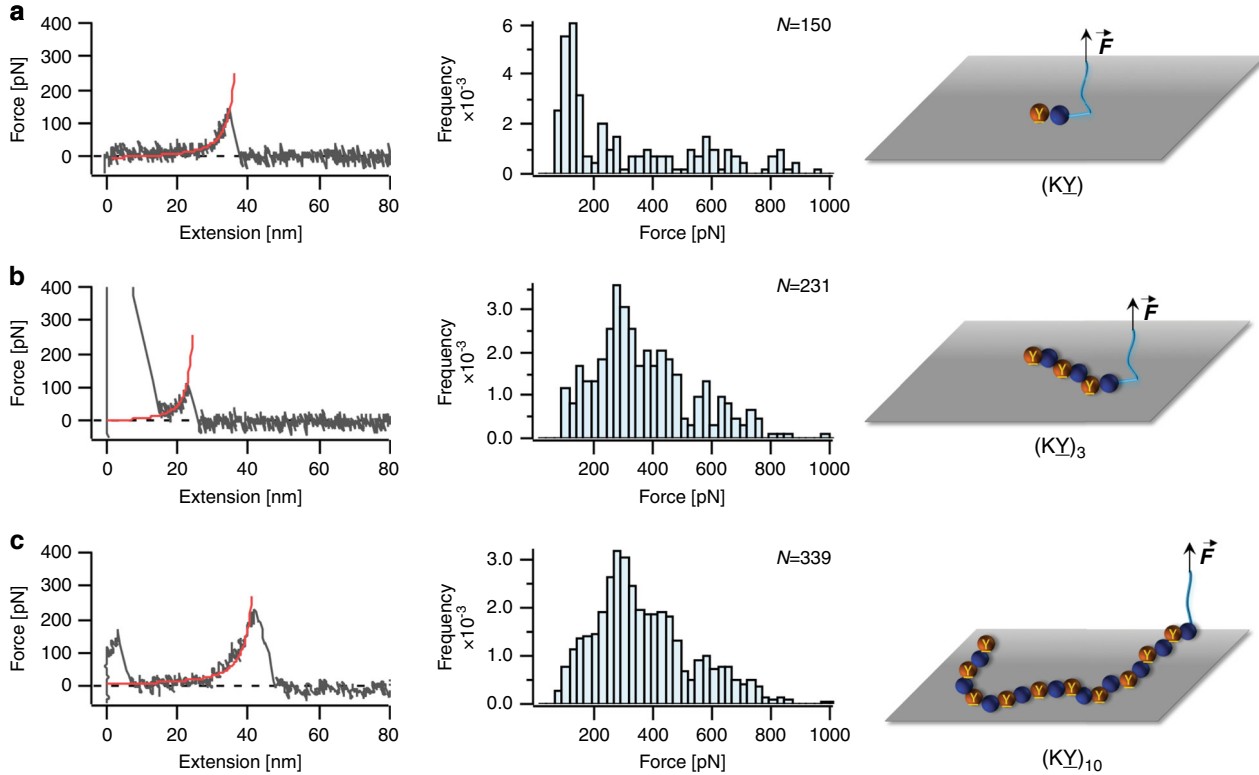

**Fig. 3 SMFS results for the interaction of (KY), (KY)₃, and (KY)₁₀ peptides with TiO₂.** Representative F–X curves (left column) and rupture force distribution (middle column) are shown for **a** (KY), **b** (KY)₃, and **c** (KY)₁₀. N values represent the total number of rupture events used to plot the histograms. The red lines on the F–X curves correspond to WLC fitting. Right column shows schematic illustration of the peptides interacting with the substrate. Source data are provided as a Source data file.

To investigate whether Phe residues could have the same effect on wet adhesion as DOPA, we designed peptides with (KY) or (KF) sequences and compared the binding strength of their interactions with polystyrene (PS) and TiO₂. The representative F–X curves and the corresponding detaching force distributions are shown in Fig. 4. The rupture force distribution showed an average detachment force of ~90 pN for the interaction of (KF) dipeptide and PS surface, while no detectable rupture force was observed for the interaction with TiO₂ surface. Previous studies have showed that DOPA can utilize hydrophobic or π–π stacking interactions with aromatic rings on the surface to bind to PS substrate[9,10]. The (KF) peptide could also bind to PS surface in a similar way through interaction of phenyl groups. Besides, Lys residues can participate in cation-π interactions with the PS surface[9,10]. As a consequence, an observable detaching force was measured for the interaction with PS. However, unlike DOPA that uses its catechol to form bidentate coordination bonds with TiO₂ surface, Phe cannot form such stable interactions with TiO₂, and the charge interaction between Lys and TiO₂ surface is not strong enough to be detected by this technique[9,10,36]. Consequently, we failed to observe detectable rupture events for the interaction of (KF) and TiO₂ surface. The (KY) peptide, on the other hand, displayed detectable interactions with both PS and TiO₂ surfaces. We observed a dominant rupture force peak located at ~100 pN for both surfaces, as well as an additional broad rupture force distribution from 200 to 900 pN for TiO₂. The results suggest that (KY) is a more versatile and stronger adhesive moiety compared to (KF), possibly owing to the broader range of interfacial adhesive mechanisms of catechols. It is important to note that we sought to isolate the adhesive interactions of single molecules with substrates, and that the vanishingly low concentration of peptide on the cantilever tip precludes the formation of intramolecular

peptide aggregates[41–43]. However, at relatively higher concentrations the (KF) incorporated peptides might form stable secondary nano-assemblies which can drastically increase the intermolecular interactions and cohesion strength, leading to larger separation forces detected in the previous SFA measurements[44,45]. We further investigated the effect of incorporating more adhesive units in the peptide backbone by synthesizing (KF)₃ and (KY)₃ sequences and measuring their interaction with TiO₂ and PS substrates using the same methodology. The representative F–X curves and rupture force distributions are shown in Supplementary Fig. 13. The average detaching force for (KF)₃ against PS surface was ~90 pN, almost the same as that of the KY peptide measured before. However, since multiple Lys residues increased the positive charges on the peptide, the binding strength with negatively charged TiO₂ surface was enhanced compared to before and a detaching force of ~100 pN was observed. For the case of (KY)₃ sequence, the average detaching force on PS surface was similar to that observed for (KY) peptide. Overall, the results indicated that although Phe can perform similar to DOPA on hydrophobic surfaces that can accommodate π–π or cation–π interactions, DOPA is a more versatile adhesive motif and has clear advantages for improving interfacial adhesion on a broader range of substrates.

Next, we studied the adhesion performance of a Lys and DOPA rich Mfp-5 segment. Mfp-5 family are interfacial proteins in the plaque and have the highest DOPA content (up to 30 mol%) among of the byssal thread proteins[46]. Although the adhesive properties of Mfp-5 and its analogs have been intensively studied by SFA measurements, the molecular detachment mechanism of individual Mfp-5 proteins is still unclear[46]. The native protein has over 74 amino acids which makes it difficult to synthesize the entire Mfp-5 chain using solid phase peptide synthesis method.

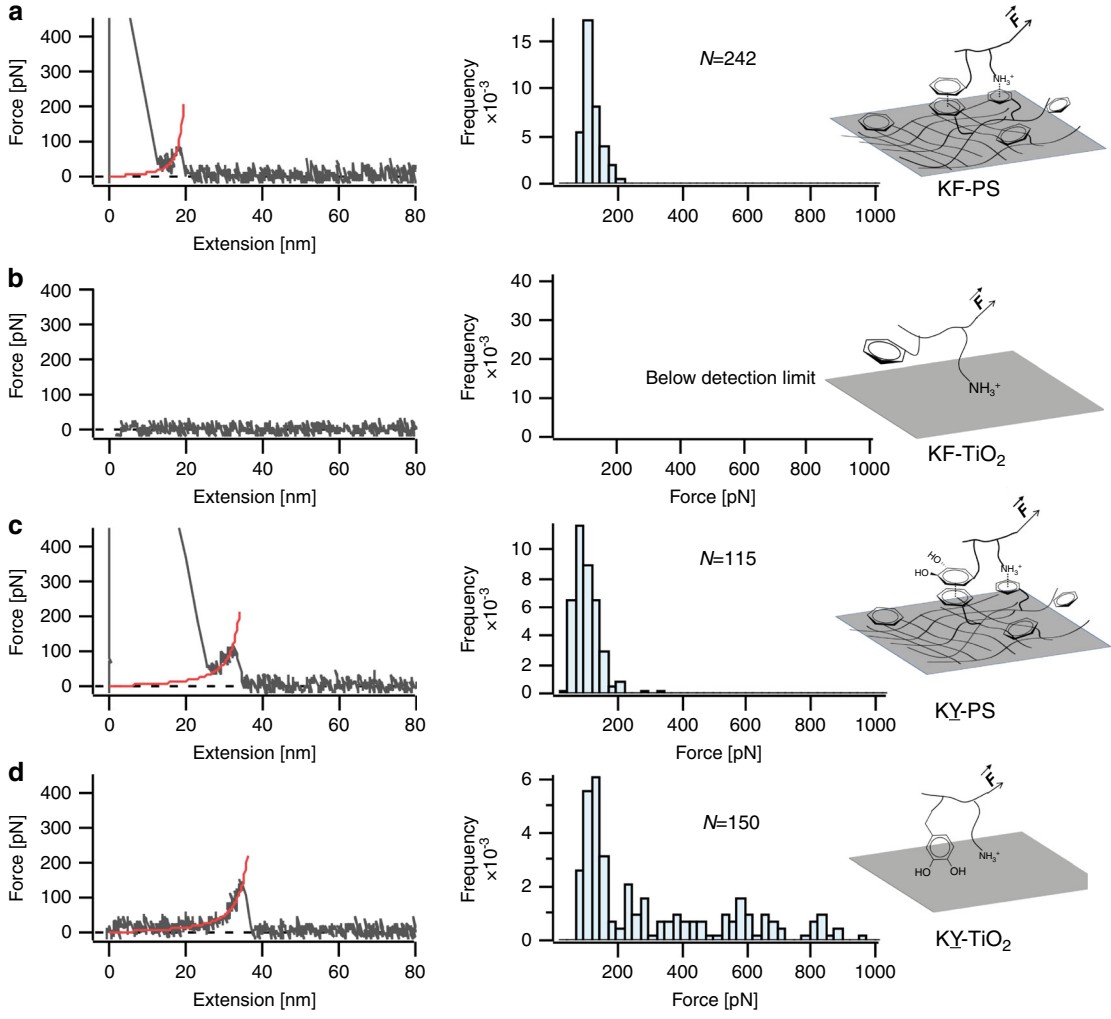

**Fig. 4 SMFS results for the interaction of (KF) and (KY) dipeptides with PS and TiO₂.** Representative F–X curves (left) and rupture force distribution (right) are shown for interaction of (KF) in **a**, **b** and for interaction of (KY) in (**c**, **d**). N values represent the total number of rupture events used to plot the histograms. The red lines in the F–X curves correspond to the WLC fitting. Schematic illustrations for the peptide–surface interactions are shown on the right. For comparison purposes, **d** is reproduced here from the same dataset as shown in Fig. 3a. Source data are provided as a Source data file.

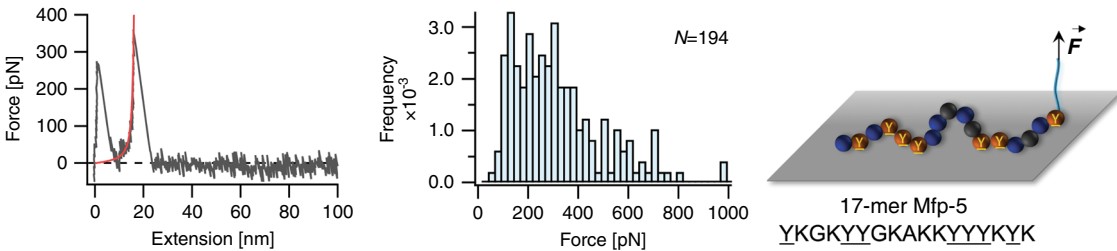

**Fig. 5 SMFS results for interaction of 17-mer Mfp-5 peptide with the TiO₂.** Representative F–X curve, rupture force distribution, and schematic of the peptide interacting with the substrate are shown from left to right, respectively. N value represents the total number of rupture events used to plot the histogram. Source data are provided as a Source data file.

Therefore, we selected a part of Mfp-5 protein with 17 amino acids and incorporated an azide functional terminus (Azide-YKGKYYGKAKKYYYKYK, and is termed as Mfp-5 analog hereafter) for coupling to the cantilever. We note that the selected segment has a higher DOPA content compared to the native Mfp-5. We then connected this Mfp-5 analog to the AFM cantilever and performed SMFS experiments as previously described for (KY) peptides. Most of the F–X curves showed a single rupture peak at ~20 nm (Fig. 5). However, in ~20% of F–X curves, we observed a sawtooth-like pattern indicating

detachment of multiple DOPA–surface interactions. The measured rupture force showed a wide distribution with a peak located at ~280 pN (Fig. 5) similar to that of the (KY)₃ sequence. The SMFS results indicated that the (KY)₃ sequence has similar adhesive performance as Mfp-5, and simply adding more adhesive units contribute little to enhancing the overall interaction strength.

Finally, a closer look at the Mfp-5 sequence reveals that several other amino acids are present and may have the effect of acting as spacers between (KY) repeat units[46]. Inspired by this design

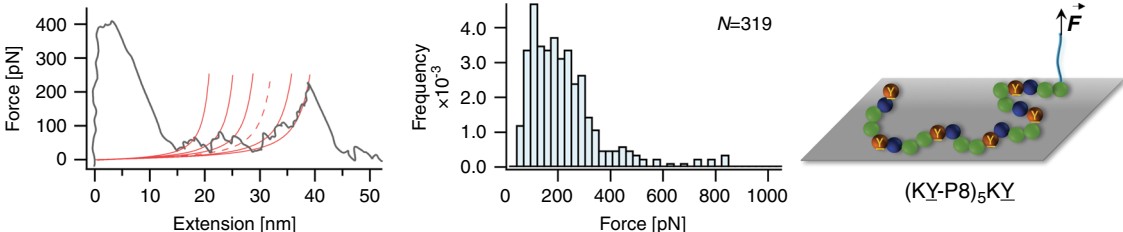

**Fig. 6 SMFS results of (KY-P8)₅KY peptide interacting with TiO₂.** Representative F–X curve and rupture force distribution for the interaction of the peptide with the surface are shown. The retraction trace is smoothed using a 10-period moving-average filter to reduce the thermal noise. N value represents the total number of rupture events used to plot the histogram. The red lines in F–X curve show WLC fittings, where dashed line corresponds to the missing rupture peak. Right column shows schematic illustration of the peptides interacting with the substrate where green spheres represent the oligoethylene glycol spacer between the KY units. Source data are provided as a Source data file.

principle and considering the probability of observing multiple rupture peaks in the F–X curves for the Mfp-5 analog was much higher than those of (KY)₃ and (KY)₁₀ sequences, we were intrigued to synthesize and test adhesion of model peptides with similar binding-site density as Mfp-5. We used monodisperse oligoethylene glycol oligomers (P8) to mimic the spacer residues in native Mfp-5 and synthesized peptides with multiple (KY-P8) repeat units (Fig. 6). Same cantilever modification method and experimental conditions as before were used to measure the interaction of the peptide with the TiO₂. Most of the F–X curves displayed 3–4 distinct rupture peaks with contour length increment of ~5 nm that is commensurate with the (KY-P8) length. The detaching force for the peptide showed a broad distribution with a peak located at ~200 pN (Fig. 6). Compared to the peptide with three adjacent (KY) units, a flexible oligoethylene glycol linker reduces the peptide rigidity, allowing for each (KY) unit to interact with the surface effectively. Although this peptide did not show an increase in the adhesion strength as measured by SMFS, the oligoethylene glycol spacer can act as hidden length and lead to dissipating more energy during the detachment process[47,48]. Unlike constructs of KY repeats with no spacing where the adhesive sites are detached almost simultaneously upon application of force, in the P8 incorporated peptide the adhesive motifs are detached sequentially followed by release of the hidden length. The energy dissipated in this process can further confer toughness and superior performance to the adhesive[47,48].

We used AFM-based SMFS technique to study the influence of length, composition and topological structure on the adhesion strength of mussel-inspired peptides. We synthesized a library of peptides with different number of Lys-DOPA repeat units as well as a 17-mer peptide from Mfp-5 and demonstrated that a modest increase in the length of the peptide can lead to noticeable increase in the adhesion strength while further increasing the peptide length without incorporating nonadhesive spacer between Lys-DOPA units resulted in little benefits on the adhesion strength. Moreover, we substituted Phe for DOPA in the peptide sequences and showed that while Lys-Phe and Lys-DOPA containing peptides could interact with organic substrates similarly, no strong interfacial adhesion could be detected for the interaction of Lys-Phe sequences against inorganic substrates, further highlighting the versatility of DOPA in establishing strong interfacial adhesion to a broader range of substrates. And finally, we designed a peptide with nonadhesive molecular spacer incorporated between Lys-DOPA repeats and showed that this hidden length can enhance the overall adhesive performance by allowing for higher energy dissipation before ultimate rupture. The findings in this work can provide a solid foundation to tailor properties and further guide the deliberate design and synthesis of bioinspired wet adhesives.

## Methods

**Materials.** Fmoc-DOPA(acetonide)-OH was purchased from Chempep Inc. N₃-P₆-COOH and Fmoc-amido-P₈-acid were purchased from Broadpharm. Dibenzocyclooctyne-amine (DBCO), Fmoc-Lys(Boc)-OH, Fmoc-Phe-OH, Rink Amino MBHA resin, N,N,N′,N′-Tetramethyl-O-(1H-benzotriazol-1-yl)uronium hexafluorophosphate (HBTU), N,N-Diisopropylethylamine (DIPEA), 1,2-Ethane-dithiol (EDT), Triisopropylsilane (TIPS), and a-cyano-4-hydroxy-cinnamic acid (CHCA) were purchased from Sigma-Aldrich. PBS powder concentrate was purchased from Fisher scientific. Sulfuric acid was from VWR and hydrogen peroxide was from Acros. Maleimide-PEG-NHS (5000 Da) was acquired from Nanocs Inc and maleimide-PEG-methoxy (2000 Da) was obtained from Jenkem.

**Solid phase peptide synthesis.** Peptides were synthesized on an automated microwave peptide synthesizer (Liberty Blue, CEM) with Fmoc-strategy (Supplementary Fig. 1). The Rink Amino MBHA resin (loading scale 0.2 mmol) was swelled in DMF for 30 min and ran through the following cycles of Fmoc-deprotection and amino acid coupling. Fmoc-deprotection: $T = 75°C$, power of microwave = 30 W, $t = 5$ min with piperidine (20 w% in DMF, 5 mL), three wash steps (DMF, 5 mL) after deprotection. Amino acid coupling: $T = 75°C$, power of microwave = 30 W, $t = 10$ min for Fmoc-Lys(Boc)-OH and 15 min for all other Fmoc-amino acid, with Fmoc-amino acid (0.2 M in DMF, 2 mL, 2 eq), HBTU (0.2 M in DMF, 2 mL, 2 eq) and DIPEA (1 M in DMF, 1 mL, 5 eq), two wash steps (DMF, 5 mL) after coupling.

**N3-P6-COOH coupling.** Peptides on resin were deprotected in 20% of piperidine/DMF at RT for 30 min and then washed with 5 mL of DMF for three times. N3-P6-COOH (0.2 M in DMF, 1.1 mL, 1.1 eq), HBTU (0.2 M in DMF, 1.1 mL, 1.1 eq) and DIPEA (0.2 M in DMF, 4 mL, 4 eq) were added into the resins and shaken for 2 h (100 rpm, 37 °C). The resins were washed by 5 mL of DMF for three times.

**Resin cleavage.** Resin cleavage was performed with a mixture of TFA/DCM/EDT/TIPS (50:45:2.5:2.5, 10 mL) for 2 h. Peptides with 6 amino acid units or more were concentrated by rotavapor and precipitated from cold diethyl ether (90 mL), washed twice with diethyl ether and dried by N₂ blowing. Peptides with two amino acid units (KF and KY) were concentrated by rotavapor and dried under high vacuum.

**Purification and characterization.** Crude peptides were dissolved in 0.1% aqueous TFA to 10 mg/mL, and injected (2 mL/run) into a semi-preparative HPLC system. Analytical HPLC curves were obtained on Agilent 1260 Infinity Quarternary LC System with Vydac 218TP C18, 10 μm, ID 4.6 × 250 mm column, 0–100% B in 30 min gradient (A: water + 0.1% TFA, B: acetonitrile + 0.1% TFA), 1.00 mL/min flow rate, and 230 nm UV detector. After purification, the yields for peptides with 2 and 6 amino acids (KY, (KY)₃, KF, (KF)₃) were ~10% (based on the ~0.7 mmol resin loading) after purification. The synthetic yields for longer chain peptides were roughly 2–3%.

Collected fractions were verified by MALDI spectra (Voyager DE Pro instrument) on reflector mode with CHCA as matrix, concentrated by rotavapor and lyophilized to obtain purified peptides. HPLC: Agilent 1260 Infinity Quarternary LC System, Column: ZORBAX SB-C18, 5 μm, ID 9.4 × 250 mm, Gradient: 0–15% B in 5 min, then 15–30% B in 20 min (A: water + 0.1% TFA, B: acetonitrile + 0.1% TFA), Flow rate: 4.00 mL/min.

**Cantilever modification.** MLCT Silicon nitride cantilever (Bruker Nano Inc.) were first treated with piranha solution (H₂O₂:H₂SO₄ = 1:5 (v:v), Piranha is a very aggressive solution and should be used with caution) for 30 min. After rinsing with excessive DI water and gently drying under a stream of nitrogen, the cantilevers were transferred into 0.5% (v/v) MPTMS/toluene solution for 2 h for thiol functionalization. The cantilevers were then rinsed with excess toluene to remove the

unreacted MPTMS and placed in oven at 120 °C for 15 minutes to cure the alkoxysilane layer. Next the cantilevers were immersed in a 1:10 mixture of maleimide-PEG-NHS (5000 Da) and maleimide-PEG-methoxy (2000 Da), at a total concentration of 1 mg ml$^{-1}$, in DMSO for 3 h. This ratio was used to control the binding density of bifunctional PEG and to reduce nonspecific interactions in the force spectroscopy measurements. The cantilevers were then rinsed with DMSO and incubated in a 0.5 mg ml$^{-1}$ solution of DBCO-amine in DMSO with 0.2% (v/v) trimethylamine for 2 h. The cantilevers were washed with DMSO to remove unreacted reagent and were incubated in PBS (10 mM phosphate, 137 mM NaCl, pH 9) for 4 h to hydrolyze maleimide-thiol bond to a stable ring-opened form. Finally, the cantilevers were immersed into 1 mg ml$^{-1}$ solution of peptides in DMSO for 1 h. The modified cantilevers were then washed with DMSO and ethanol then dried under a stream of nitrogen.

**Substrate preparation**. TiO$_2$ substrates were treated with Piranha solution for 30 min to remove organic residues from the surface. The substrates were then rinsed extensively with water and dried with nitrogen. Polystyrene substrates were cleaned by sonication in ethanol for 1 h and then rinsed with excessive water and dried with nitrogen.

**AFM-based force spectroscopy measurements**. AFM force spectroscopy measurements were carried out using a JPK ForceRobot 300 AFM (JPK Instruments AG, Germany). The experiments were performed in 10 mM PBS buffer (containing 137 mM NaCl) which was previously bubbled with nitrogen to degas dissolved oxygen to minimize catechol oxidation during the course of measurements. However, since the AFM chamber was exposed to air during force curve collection, it is expected that some oxygen was present in the buffer. Soft silicon nitride MLCT cantilevers of typical spring constant of 50–60 pN nm$^{-1}$ were used for all experiments and calibrated using the thermal tune method after allowing the cantilever to equilibrate in solution for at least 30 min[49]. In a typical force measurement, the cantilever was approached to substrate at a constant speed of 1000 nm s$^{-1}$ and held at the surface for 2 s to allow for the interaction between peptides and substrates. The cantilever was then retracted at the same speed. The force-extension curves were recorded using JPK data processing software and were further analyzed by a custom-written procedure in Igor Pro 6.12 (Wavemetric, Inc).

## Data availability

The source data underlying Figs. 3–6 and Supplementary Figs. 9–13 are provided as Source data file with this paper and are available online at https://doi.org/10.6084/m9.figshare.12502163.v1. Other data are available from the corresponding author upon reasonable request. Source data are provided with this paper.

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

## Acknowledgements

This work was supported by National Institute of Health (NIH) grants R37 DE014193 and R01EB022031, and the National Natural Science Foundation of China (No. 11674153). S.J.S was supported by Swiss National Science Foundation Early PostDocMobility Fellowship (No. P2BSP2_168718). K.G.D would like to acknowledge the National Science Foundation Graduate Research Fellowship (No. DGE 1752814).

## Author contributions

Y.L., J.C., P.D., and H.W. contributed equally to this work. Y.L., P.D., S.J.S., J.C., and P.B.M. planned the experiments. Y.L., P.D., and H.W. performed all AFM cantilever modification and SMFS experiments. Y.L., P.D., and H.W. analyzed the SMFS results. J.C., S.J.S., and K.G.D. synthesized and characterized the peptides. Finally, Y.L., P.D., Y.C. and P.B.M. wrote and revised the paper. All authors have given approval to the final version of the paper.

## Competing interests

The authors declare no competing interests.
