## [Peer Review File · Nature Communications]

Reviewers' Comments:

Reviewer #1:

Remarks to the Author:

This manuscript, entitled "Topological Design Principles of Lysine-DOPA Wet Adhesion: Single-Molecule Perspective", reports an experimental study of the influence of chemical composition and structure on the adhesion properties of mussel-inspired peptides. Studied results show that as the number of repeating structural units increase within a specific range, with the peptide number from KY to (KY)3, the adhesion strength to TiO₂ increases significantly, and with the further increase of peptide chain structural units, increase in peptide length to (KY)10, it does not show an additional adhesive increase. The effect of different adhesive moieties and molecular spacer incorporated between repeated units on the adhesion strength was also explored. The method provided in this work for exploring the molecular adhesion behavior is both convenient and practical. The selected system is interesting and some novel results got have a considerable practical significance. The present manuscript is suitable for publication in Nature Communications after a minor revision as pointed bellow.

Comments

For the case of (KY)10 peptide, the electrostatic repulsion between the repeating structural units (KY units) would likely affect the simultaneous interaction of each structural unit with the substrate surface, for which the author should give further discussion or explanation.

Due to the presence of spacer groups (PEG spacer), the strength of the interaction is greatly affected. However, it is necessary to further explain why this group can have such a great influence, and make further explanation from the perspective of interaction force or steric hindrance effect. For example, the presence of spacer can reduce the electrostatic repulsion between repeating structural units (KY), thereby affecting the interaction behavior of these structural units and the substrate surface, etc..

Reviewer #2:

Remarks to the Author:

Recommendation: Minor corrections

Comments:

This manuscript by Messersmith and coworkers presents a single-molecule force spectroscopy (SMFS) study on the adhesion strength of mussel-inspired peptides to elucidate the effect of molecular topology over wet adhesion. Inspired by the sequence of the adhesive mussel foot protein 5 (mfp-5) that contains DOPA and Lys moieties, a library of peptides containing adhesive dipeptide units of Lys and either DOPA (KY) or Phe (KF) were synthesized with variable length, number of repeating units (1, 3 or 10), amino acids order in the dipeptide (KY vs YK) and presence of a non-adhesive and flexible oligoethylenglycol spacer between the repeating units. These peptides were covalently bound to an AFM cantilever and probed via SMFS under wet conditions (PBS) to two different surfaces: a hydrophilic charged inorganic surface (TiO₂) and an organic hydrophobic substrate (polystyrene). The adhesion strength of the different peptide systems was measured and compared to reveal the effect of the molecular design over adhesion performance. Additionally, a 17-mer mimic of mfp-5 was synthesized as well and used for comparison.

The authors found that: i) increasing the number of repeat units from 1 to 3 results in higher adhesion, while further increase to 10 does not noticeably enhance adhesion. ii) KY and KF derivatives showed similar adhesion to the hydrophobic polystyrene, while KY showed much higher adhesion onto TiO₂. This was attributed to the molecular ability of Y to establish more and diverse interfacial adhesive interactions with surfaces of different kinds. This result points to a higher versatility of Y vs F as a functional component of adhesive formulations designed to perform over various substrates. iii) The incorporation of a flexible, non-adhesive spacer was found to reduce

the overall peptide rigidity and allowed each KY unit to interact with the surface more efficiently. Although in this particular case the adhesion strength did not increment, the inclusion of a spacer provided a hidden length to the structure that could be useful to increase energy dissipation before rupture, which would ultimately enhance the adhesive performance of a formulation.

The novelty of this study is somehow limited. However, the findings of this work are highly relevant to guide the rational molecular design of mussel inspired underwater adhesives and are interesting to the community of adhesive and bioinspired materials. Therefore, I rather support its publication. The article is well presented, is easy to follow and offers an adequate discussion in relation to the current literature.

I recommend the following minor corrections:

1. Page 1, abstract, line 2: "DOPA" should be defined at first mention.
2. Page 2, line 23: "SFMS" should be defined at first mention.
3. Page 4, lines 7-8: When mentioning the library of peptides synthesized, please, include a sentence to refer to Table 1.
4. Page 4, lines 20-23: in the part related to the synthesis of the peptide library, what is the yield of such syntheses? No data is currently included about this point, neither in the main manuscript nor in the SI. Please, comment on that.
5. Pages 4-5: when describing a typical force spectroscopy experiment, could you please mention what is the occurrence rate (in %) of observation of the rupture events? I would expect this to be quite low.
6. Page 10, lines 10-12: "We used monodisperse PEG oligomers (P8) to mimic the 'spacer' residues". To my point of view, describing the spacer as "PEG oligomers" is not accurate. It is recommended that this term is replaced by just "oligoethylene glycol spacer".
7. Page 10, lines 16-17: in relation to the statement "The energy dissipated in this process can further confer toughness and superior performance to the adhesive", I recommend to add references to back this up.
8. Page 19, caption of Table 1: "Calculated and measured molecular weight values (obtained from MALDI) are shown". This is not accurate. Mass spec techniques provide information on mass-to-charge ratio, where the "mass" refers to "exact mass" and not to molecular weight. Therefore, the correct terms should be "calculated exact mass" and "measured or found mass". The same correction should be introduced to the caption of Figure S1 at the SI.

Reviewer #3:

Remarks to the Author:

Review of "Topological Design Principles of Lysine-DOPA Wet Adhesion: Single-Molecule Perspective"

This is an outstanding communication that reports single molecule force spectroscopy on a series of DOPA-Lysine peptides with comparison to Phe-Lys peptides. The paper demonstrates the synergistic effect of DOPA and Lys in wet adhesion to TiO₂ and shows that increasing the length of the peptide increases adhesion strength, up to a point, but more intriguing is incorporating a region of nonadhesive spacing between Lys-DOPA units also increases adhesion. These are new, important and exciting results, that increase our understanding of mussel foot protein adhesion in significant ways. This communication differs substantially from a previous report in Science on the synergistic relationship between catechol and primary amines in aqueous adhesion, in that the current results use peptides with DOPA as opposed to a different catechol on a non peptidic base, and this communication uses SMFS to investigate adhesion to TiO₂ versus the Science report using SFA on mica. The authors suggest that their work provides a solid foundation guiding the deliberate design of bioinspired wet adhesives, however they fail to bring out in the main text specifically the fact that their experiments are all carried out under anaerobic conditions to prevent

oxidation of DOPA at neutral pH under their PBS conditions – This conclusion, as written, seems somewhat overstated – since wet adhesive materials would not likely be especially useful in only anaerobic environments. However, overall, I find this is well written communication, and I recommend that it be accepted with only very minor revisions.

Since the experiments are carried out at neutral pH in PBS and since the manuscript indicates catechol interaction with TiO₂ is via coordination to Ti(IV), what happens to the SMFS of these compounds at lower pH? Also one wonders how the adhesive properties of Lys-Tyr compare to Lys-DOPA and Lys-Phe – but these are not required to be addressed in this communication

Minor points:

In the Abstract, writing “(DOPA, Y)” is misleading since Y is for Tyrosine – Maybe HO^(superscript)Y would be better to indicate hydroxylated Tyr – and elsewhere in the text.

Page 7, define PS

P 8 line 13 the word ‘the’ is missing: “almost the same”

Conclusion: reword on of the two successive sentences beginning with “Moreover,”

Authors' Rebuttal to Reviewers' Comments:

Reviewer: 1

Recommendation: Publish after a minor revision noted.

Comments:

This manuscript, entitled “Topological Design Principles of Lysine-DOPA Wet Adhesion: Single-Molecule Perspective”, reports an experimental study of the influence of chemical composition and structure on the adhesion properties of mussel-inspired peptides. Studied results show that as the number of repeating structural units increase within a specific range, with the peptide number from KY to $(KY)_3$, the adhesion strength to TiO_2 increases significantly, and with the further increase of peptide chain structural units, increase in peptide length to $(KY)_{10}$, it does not show an additional adhesive increase. The effect of different adhesive moieties and molecular spacer incorporated between repeated units on the adhesion strength was also explored.

The method provided in this work for exploring the molecular adhesion behavior is both convenient and practical. The selected system is interesting and some novel results got have a considerable practical significance. The present manuscript is suitable for publication in Nature Communications after a minor revision as pointed bellow.

1. For the case of $(KY)_{10}$ peptide, the electrostatic repulsion between the repeating structural units (KY units) would likely affect the simultaneous interaction of each structural unit with the substrate surface, for which the author should give further discussion or explanation. Due to the presence of spacer groups (PEG spacer), the strength of the interaction is greatly affected. However, it is necessary to further explain why this group can have such a great influence, and make further explanation from the perspective of interaction force or steric hindrance effect. For example, the presence of spacer can reduce the electrostatic repulsion between repeating structural units (KY), thereby affecting the interaction behavior of these structural units and the substrate surface, etc..

RESPONSE:

We thank you for your comments about the manuscript. We indeed agree with your comment regarding the role of electrostatic repulsion between the repeat units on the observed adhesion. We previously described the effects of peptide rigidity and conformation briefly in the beginning lines of page 7. We have revised this section based on your suggestion to clarify and provide more information on the role of electrostatic interactions.

“For the case of $(KY)_{10}$ peptide, surface roughness as well as backbone rigidity and peptide conformation due to increased electrostatic repulsion between positive charges of Lys groups might possibly make it highly unlikely for all the (KY) units to simultaneously interact with the substrate effectively. As a result, the adhesion strength is not significantly enhanced by simply incorporating more KY repeat units in the peptide.”

Regarding the effects of incorporating a flexible spacer between the adhesive KY units, as described in page 10 and Figure 5, the results do not show an increase in the adhesion strength. Presence of a flexible linker can possibly lead to reducing the electrostatic repulsion between the repeat units and result in a less rigid backbone that can more effectively interact with the substrate, compared to (KY)₃ and (KY)₁₀ peptides where the adhesive units are positioned adjacent to each other. We have revised the corresponding section in page 10 to further highlight the role of the flexible linker in reducing electrostatic repulsion forces as well as steric hindrance of the peptide.

“Compared to the peptide with 3 adjacent (KY) units, a flexible oligoethylene glycol linker reduces the electrostatic repulsion forces as well as steric hindrance and peptide rigidity, allowing for each (KY) unit to interact with the surface effectively. Although this peptide did not show an increase in adhesion strength as measured by SMFS, the oligoethylene glycol spacer can act as hidden length and lead to dissipating more energy during the detachment process.^{1,2} Unlike constructs of KY repeats with no spacing where the adhesive sites are detached almost simultaneously upon application of force, in the P8 incorporated peptide the adhesive motifs are detached sequentially followed by release of the hidden length. The energy dissipated in this process can further confer toughness and superior performance to the adhesive.”

Reviewer: 2

Recommendation: Publish after minor corrections noted.

Comments:

This manuscript by Messersmith and coworkers presents a single-molecule force spectroscopy (SMFS) study on the adhesion strength of mussel-inspired peptides to elucidate the effect of molecular topology over wet adhesion. Inspired by the sequence of the adhesive mussel foot protein 5 (mfp-5) that contains DOPA and Lys moieties, a library of peptides containing adhesive dipeptide units of Lys and either DOPA (KY) or Phe (KF) were synthesized with variable length, number of repeating units (1, 3 or 10), amino acids order in the dipeptide (KY vs YK) and presence of a non-adhesive and flexible oligoethyleneglycol spacer between the repeating units. These peptides were covalently bound to an AFM cantilever and probed via SMFS under wet conditions (PBS) to two different surfaces: a hydrophilic charged inorganic surface (TiO₂) and an organic hydrophobic substrate (polystyrene). The adhesion strength of the different peptide systems was measured and compared to reveal the effect of the molecular design over adhesion performance. Additionally, a 17-mer mimic of mfp-5 was synthesized as well and used for comparison.

The authors found that: i) increasing the number of repeat units from 1 to 3 results in higher adhesion, while further increase to 10 does not noticeably enhance adhesion. ii) KY and KF derivatives showed similar adhesion to the hydrophobic polystyrene, while KY showed much higher adhesion onto TiO₂. This was attributed to the molecular ability of Y to establish more and diverse interfacial adhesive interactions with surfaces of different kinds. This result points to a higher versatility of Y vs F as a functional component of adhesive formulations designed to

perform over various substrates. iii) The incorporation of a flexible, non-adhesive spacer was found to reduce the overall peptide rigidity and allowed each KY unit to interact with the surface more efficiently. Although in this particular case the adhesion strength did not increment, the inclusion of a spacer provided a hidden length to the structure that could be useful to increase energy dissipation before rupture, which would ultimately enhance the adhesive performance of a formulation.

The novelty of this study is somehow limited. However, the findings of this work are highly relevant to guide the rational molecular design of mussel inspired underwater adhesives and are interesting to the community of adhesive and bioinspired materials. Therefore, I rather support its publication. The article is well presented, is easy to follow and offers an adequate discussion in relation to the current literature.

I recommend the following minor corrections:

RESPONSE:

We thank the reviewer for the feedback and suggested corrections to improve the manuscript. Our point-by-point response to the comments are below.

1. Page 1, abstract, line 2: “DOPA” should be defined at first mention.

Thank you for bringing this to our attention. We have added definition of DOPA as 3,4-dihydroxyphenylalanine to line 2 of Abstract.

2. Page 2, line 23: “SMFS” should be defined at first mention.

Thank you for pointing this out. SMFS has been previously defined in lines 11-12 of Abstract (where it has first been mentioned) as Single Molecule Force Spectroscopy.

3. Page 4, lines 7-8: When mentioning the library of peptides synthesized, please, include a sentence to refer to Table 1.

A reference to Table 1 has been added to the sentence as suggested by the comment.

4. Page 4, lines 20-23: in the part related to the synthesis of the peptide library, what is the yield of such syntheses? No data is currently included about this point, neither in the main manuscript nor in the SI. Please, comment on that.

Thank you for pointing this out. The yield for peptides with 2 and 6 amino acids (KY, (KY)₃, KF, (KF)₃) was ~10% after purification. The synthetic yields for longer chain peptides were roughly 2-3%.

We have updated the methods section in the Supplementary Materials to reflect this information.

5. Pages 4-5: when describing a typical force spectroscopy experiment, could you please mention what is the occurrence rate (in %) of observation of the rupture events? I would expect this to be quite low.

In general, the observation rate for rupture events in force spectroscopy measurements are highly dependent on experimental conditions such as contact force and dwell time as well as surface properties and cantilever modification. For a typical measurement on average ~1-5% of the total collected force-distance curves contain single-molecule rupture features.

6. Page 10, lines 10-12: “We used monodisperse PEG oligomers (P8) to mimic the ‘spacer’ residues”. To my point of view, describing the spacer as “PEG oligomers” is not accurate. It is recommended that this term is replaced by just “oligoethylene glycol spacer”.

Thank you for your suggestion. We agree with your comment and thus have substituted mentions of PEG on page 10 with oligoethylene glycol oligomers.

7. Page 10, lines 16-17: in relation to the statement “The energy dissipated in this process can further confer toughness and superior performance to the adhesive”, I recommend to add references to back this up.

We have added the following references to the mentioned sentence:

- Smith, B., Schäffer, T., Viani, M. et al. Molecular mechanistic origin of the toughness of natural adhesives, fibres and composites. *Nature* 399, 761–763 (1999).
- Fantner, G., Hassenkam, T., Kindt, J. et al. Sacrificial bonds and hidden length dissipate energy as mineralized fibrils separate during bone fracture. *Nature Mater* 4, 612–616 (2005).

8. Page 19, caption of Table 1: “Calculated and measured molecular weight values (obtained from MALDI) are shown”. This is not accurate. Mass spec techniques provide information on mass-to-charge ratio, where the “mass” refers to “exact mass” and not to molecular weight. Therefore, the correct terms should be “calculated exact mass” and “measured or found mass”. The same correction should be introduced to the caption of Figure S1 at the SI.

We agree with your comment and have revised Table 1 and Figure S1 captions accordingly.

Reviewer: 3

Recommendation: Accepted with very minor revisions as noted.

Comments:

This is an outstanding communication that reports single molecule force spectroscopy on a series of DOPA-Lysine peptides with comparison to Phe-Lys peptides. The paper demonstrates the synergistic effect of DOPA and Lys in wet adhesion to TiO₂ and shows that increasing the length

of the peptide increases adhesion strength, up to a point, but more intriguing is incorporating a region of nonadhesive spacing between Lys-DOPA units also increases adhesion. These are new, important and exciting results, that increase our understanding of mussel foot protein adhesion in significant ways. This communication differs substantially from a previous report in *Science* on the synergistic relationship between catechol and primary amines in aqueous adhesion, in that the current results use peptides with DOPA as opposed to a different catechol on a non peptidic base, and this communication uses SMFS to investigate adhesion to TiO_2 versus the *Science* report using SFA on mica. The authors suggest that their work provides a solid foundation guiding the deliberate design of bioinspired wet adhesives, however they fail to bring out in the main text specifically the fact that their experiments are all carried out under anaerobic conditions to prevent oxidation of DOPA at neutral pH under their PBS conditions – This conclusion, as written, seems somewhat overstated – since wet adhesive materials would not likely be especially useful in only anaerobic environments. However, overall, I find this is well written communication, and I recommend that it be accepted with only very minor revisions.

RESPONSE:

We thank the reviewer for the feedback and suggested corrections to improve the manuscript. Our point-by-point response to the comments are below.

The authors suggest that their work provides a solid foundation guiding the deliberate design of bioinspired wet adhesives, however they fail to bring out in the main text specifically the fact that their experiments are all carried out under anaerobic conditions to prevent oxidation of DOPA at neutral pH under their PBS conditions – This conclusion, as written, seems somewhat overstated – since wet adhesive materials would not likely be especially useful in only anaerobic environments.

Actually, the experiments were performed in an aqueous buffer without exclusion of oxygen. The purpose of purging the buffer with nitrogen before beginning the experiment was to minimize changes to the catechol by oxidation during the long period of time necessary to collect several thousands of force curves. However, since the AFM chamber was open to air during measurement we do not consider this an anaerobic condition. We have added a comment to the methods section to clarify this.

1. Since the experiments are carried out at neutral pH in PBS and since the manuscript indicates catechol interaction with TiO_2 is via coordination to Ti(IV) , what happens to the SMFS of these compounds at lower pH? Also one wonders how the adhesive properties of Lys-Tyr compare to Lys-DOPA and Lys-Phe – but these are not required to be addressed in this communication

As described briefly in this work and in more detail in many comprehensive review articles on adhesive properties of DOPA, catechol group can interact with metal oxide substrates through a myriad of interfacial interactions including hydrogen bonds (with one or both hydroxyl groups) as well as coordination complexes (Figure 1 below).^{3,4} Since each SMFS experiment involves the collection and analysis of a large number of force-displacement curves, we decided to focus on the effects of chemical and topological structure of the peptide on the interfacial adhesion and

did not explore the effects of pH on the adhesion strength. However, we believe it is possible to observe hydrogen bonds as well as single coordination interactions at lower pH in SMFS.

Figure 1. Interfacial catechol bonding to metal oxide surfaces changes from H-bonds at acid pH to bidentate coordination at basic pH. Adapted from Mussel adhesion – essential footwork, J. Herbert Waite, *Journal of Experimental Biology* 2017 220: 517-530; doi: 10.1242/jeb.134056

Regarding the interfacial adhesive properties of Tyr residues, although it has been previously shown in SMFS experiments that Tyr is not as adhesive as DOPA,⁵ we agree with your suggestion and believe that there are certainly opportunities to further investigate the possible synergistic effects between Lys and Tyr in other works. We think this article can motivate further research in this area.

Minor points

2) In the Abstract, writing “(DOPA, Y)” is misleading since Y is for Tyrosine – Maybe HO^(superscript)Y would be better to indicate hydroxylated Tyr – and elsewhere in the text.

We have further defined DOPA as 3,4-dihydroxyphenylalanine in the abstract and throughout the manuscript we used Y (with an underline) to denote DOPA residues to avoid any confusion with Tyr. We followed the conventional nomenclature in the mussel adhesion literature for the peptide notations in order to be consistent with other publications.^{3,6-8} Moreover, since Tyr has not been mentioned in the manuscript we think the current notation should not lead to confusion between DOPA and Tyr in the text.

3) Page 7, define PS

Polystyrene has been defined on page 7 as suggested.

4) Page 8 line 13 the word ‘the’ is missing: “almost the same”

Conclusion: reword on of the two successive sentences beginning with “Moreover,”

Both sentences have been revised accordingly.

REFERENCES

- 1 Fantner, G. E. *et al.* Sacrificial bonds and hidden length dissipate energy as mineralized fibrils separate during bone fracture. *Nat Mater* **4**, 612-616, doi:10.1038/nmat1428 (2005).
- 2 Smith, B. L. *et al.* Molecular mechanistic origin of the toughness of natural adhesives, fibres and composites. *Nature* **399**, 761, doi:10.1038/21607 (1999).
- 3 Waite, J. H. Mussel adhesion - essential footwork. *J Exp Biol* **220**, 517-530, doi:10.1242/jeb.134056 (2017).
- 4 Yu, J. *et al.* Adhesion of Mussel Foot Protein-3 to TiO₂ Surfaces: the Effect of pH. *Biomacromolecules* **14**, 1072-1077, doi:10.1021/bm301908y (2013).
- 5 Das, P. & Reches, M. Revealing the role of catechol moieties in the interactions between peptides and inorganic surfaces. *Nanoscale* **8**, 15309-15316 (2016).
- 6 Lee, H., Scherer, N. F. & Messersmith, P. B. Single-molecule mechanics of mussel adhesion. *P Natl Acad Sci USA* **103**, 12999-13003 (2006).
- 7 DeMartini, D. G., Errico, J. M., Sjoestroem, S., Fenster, A. & Waite, J. H. A cohort of new adhesive proteins identified from transcriptomic analysis of mussel foot glands. *J R Soc Interface* **14** (2017).
- 8 Lee, B. P., Messersmith, P. B., Israelachvili, J. N. & Waite, J. H. Mussel-Inspired Adhesives and Coatings. *Annu Rev Mater Res* **41**, 99-132 (2011).